# Species abundances surpass richness effects in the biodiversity-ecosystem function relationship across marine fishes

Helen F. Yan [1,2] ✉, Renato A. Morais [3] & David R. Bellwood [1]

High biological diversity (or biodiversity) is thought to bolster communities against disturbances, leading to higher levels of ecosystem functioning. While the biodiversity-ecosystem function (BEF) relationship is evident in studies equating diversity to species richness, it is still unclear which ecological mechanisms can produce different observational BEF effects. Here, we combine 7686 individual growth curves across 1480 species with 2957 local community surveys to generate a process-based estimate of biomass production to assess the BEF relationship across marine reef fishes. We find that the effects of Hill diversity emphasising abundances outpace those of species richness and community evenness on biomass productivity. In high-latitude temperate regions, species abundances and richness have parallel effects on reef fish productivity. However, in the tropics, species abundances surpass species richness in their effects on functioning. These latitudinal disparities can be explained by trade-offs in the relationship between abundance and per-capita productivity. As whole-community productivity remains relatively stable across most of the diversity gradient, these trade-offs are presumably driven by metabolic constraints on growth and body size imposed by warmer temperatures. It appears that biodiversity can only support ecosystem functioning to a limited extent and environmental stressors likely limit the biomass production of marine fishes globally.

With increasing anthropogenic threats to biodiversity[1,2], a pervasive question that has arisen is whether natural communities possess internal regulatory mechanisms that can bolster critical ecosystem functions. Grounded in ecological theory, the biodiversity-ecosystem function relationship, or BEF, is posited as a positive causal link between biodiversity and ecosystem functioning through complementary resource use via coexistence and niche partitioning (i.e., functional complementarity)[3–9]. While there is a general consensus that biodiversity has a positive, albeit saturating, relationship with ecosystem functioning (e.g., ref. 10), these effects appear to be sensitive to the choice of function (e.g., ref. 11), the choice of diversity

metric (e.g., Simpson Index, Shannon Entropy, or richness[12] vs. taxonomic, functional, or evolutionary, e.g., ref. 13), or the choice of analytical method (e.g., ref. 14). Indeed, scientists scrutinise nearly every axis of the BEF, yet there is one critical factor that influences community assembly that is widely overlooked: species abundances[15,16] (but see refs. 17–19). In an attempt to isolate the role of biodiversity on ecosystem functioning in experimental studies, scientists artificially control for the role of abundance[20], which typically do not reflect naturally occurring abundances in the field. Across observational BEF studies, abundances are typically consolidated within the "function" component[21], such as the calculation of standing stock biomass (e.g.,

[1]Research Hub for Coral Reef Ecosystem Functions, College of Science and Engineering, James Cook University, Townsville, QLD, Australia. [2]Thriving Oceans Research Hub, School of Geosciences, University of Sydney, Camperdown, NSW, Australia. [3]Paris Sciences et Lettres Université, École Pratique des Hautes Études, EPHE-UPVD-CNRS, USR 3278 CRIOBE, Perpignan, France. ✉e-mail: helen.yan@sydney.edu.au

ref. 22), or standardised against during analyses (e.g., refs. 23,24). Dominance, which indirectly considers species relative abundances, is used in observational BEF studies, however, it can have variable effects across both functions (e.g., ref. 25) and ecosystems (e.g., refs. 17,18,26). While commonly used biodiversity metrics (e.g., species richness) and indices (e.g., Shannon, Simpson), and coverage-based sampling can mitigate differences in rarity and detection biases[27,28], they are unable to eliminate the effects arising from species relative abundances[12]. Indeed, experimental studies highlight that relative abundances can sway the direction of BEF studies[14,29,30] with higher abundances typically translating into higher function delivery (e.g., refs. 31–36). At the other extreme, no abundance necessarily means no function. Consequently, species' abundances may provide a valuable mechanistic link between biodiversity and function delivery.

Here, we use one of the world's largest standardised global survey programs on marine fishes (the Reef Life Survey program[37,38]), to assess whether species' abundances would influence the BEF globally. Marine fishes exhibit extraordinary patterns of taxonomic and life-history trait diversity[35,39], while contributing to a range of ecosystem functions[25,40] and providing valuable ecosystem services via fisheries. In particular, fish biomass production is one of the few key processed-based ecosystem functions[40–42], that can be readily estimated by using mechanistic relationships (i.e., somatic growth models) with direct ramifications for fisheries[43,44]. Here, productivity is defined as the biomass accumulated via ontogenetic growth of all individuals (without mortality) over the survey area over the course of one day[40] (see Methods). By building predictive models from 7686 growth curves across 1480 species, we are able to estimate the local per-capita productivity of marine fishes from 2957 sites comprising coral and rocky reefs, spanning tropical to polar locations (Fig. S1). In doing so, we are able to (1) disentangle the role of species' abundances on the total community biomass production of marine fishes and (2) assess the role of biodiversity both globally and across geographic regions.

## Results and discussion
### Abundances skew global BEF
Biodiversity encapsulates multiple intrinsic characteristics of biological communities, including species richness and abundance[45]. Here, we used Hill diversity[46,47] as a means to quantify biodiversity and evaluate different diversity indices in explaining variation in biomass production. In short, Hill diversity is a general measure of species diversity, which can be scaled to calculate common diversity indices (e.g., Simpson Index, Shannon Entropy), but allows all measures to be expressed using the same units (i.e., units of species)[12]. Unlike other diversity metrics, changes in Hill diversity values intuitively reflect changes in the community; for example, the loss of species in a community would proportionally scale with a decrease in Hill diversity[12]. Because Hill diversity is a generalised equation, we can produce commonly used diversity indices within a single equation by varying the scaling parameter $\ell$ within a single equation (see Methods)[12,48]. By changing the scaling parameter $\ell$, we can specifically emphasise common species (e.g., Simpson Index, Shannon entropy) or rare species (e.g., abundance effects) more than richness, thereby placing communities along an evenness-rarity continuum[12,48].

We explicitly tested four diversity indices by varying $\ell$ to generate the inverse Simpson index (which emphasises relative abundances and, thus, common species; $\ell = -1$), the exponentiated Shannon entropy ($\ell = 0$), richness effects (which emphasises rare species by considering all species equally; $\ell = 1$), and abundance effects ($\ell = 10$, the maximum value used by ref. 48; see Methods). All models were similarly structured, but with different specifications of $\ell$ used to quantify biodiversity. While all four diversity indices were relatively correlated and loaded positively on the first axis of a principal component analysis (PC1), which explained 68.4% of the total variation of the data (i.e., a matrix of productivity by sites [rows] and diversity

metrics [columns]; Fig. 1a; Table S1), increasing $\ell$ from −1 to 10 resulted in a shift in strength and magnitude of the BEF relationship (mirroring those found by ref. 48). BEF relationships varied from negative when using the inverted Simpson index ($\ell = -1$; median coefficient [90% credible interval]: −0.11 [−0.13 to −0.09]) and the exponentiated Shannon entropy ($\ell = 0$; −0.11 [−0.13 to −0.08]), to positive with richness effects ($\ell = 1$; 1.05 [1.01 to 1.08]) and abundance effects ($\ell = 10$; 1.10 [1.09 to 1.12]; Fig. 1b). The abundance index was the least correlated with all other Hill diversity metrics and PC1 (Fig. S2) and, based on leave-one-out information criterion (i.e., information criterion using a cross-validation procedure; LOOIC), generated the best-fitting BEF relationship (Table S2). We therefore used Hill diversity emphasising abundance effects as the measure of biodiversity in all subsequent analyses unless otherwise stated (hereon referred to as "diversity").

We found a decelerating relationship between diversity and log-scale community-level biomass production (Fig. 2). Despite generally positive global BEF effects, the magnitude of effects across geographic regions (i.e., tropical vs temperate regions) differed. Indeed, the BEF was strongest in the tropics (1.16 [1.12 to 1.19]) and relatively flatter in temperate regions (1.08 [1.06 to 1.11]; Fig. 2), indicating that species abundances had a greater impact on community biomass production in warm-water environments compared to cooler, high-latitude locations. While the BEF emphasising abundance effects was the best-fitting model based on LOOIC, we also compared the magnitude of abundance- versus richness-emphasised BEF effects because species richness is still a dominant component of biodiversity[46,47] and produced the second-best-fitting model (Table S2).

The impacts of richness and abundance BEF effects varied across tropical and temperate regions. Regardless of the index used, BEF effects in temperate regions produced nearly identical results (abundance index: 1.08 [1.06 to 1.11]; richness index: 1.08 [1.04 to 1.12]; Fig. 2), which suggests that species richness and abundances may have analogous impacts on community biomass production in temperate regions. The impact of species abundances and richness on fish biomass production in tropical regions, however, differed. While both BEF effects were positive, richness effects were substantially weaker than abundance effects (0.96 [0.91 to 1.02] and 1.16 [1.12 to 1.19], respectively; Fig. 2). Namely, increases in abundance-emphasised diversity correlated with greater increases in productivity than increases in species richness-emphasised diversity, which suggests that species abundances, not richness, may be the primary community-level feature giving rise to/or the consequence of highly productive tropical fish communities. Indeed, many of the richest tropical communities are characterised by small-bodied fishes occurring in higher abundances[49], while the disproportionately high abundances of planktivorous fishes, which capitalise on external pelagic subsidies, better correlated with higher community productivity than planktivore richness[32]. Although we cannot discount the role of species richness as a correlate of productivity, it appears that the iconic productivity of high-diversity systems in the tropics are predominantly driven by species abundances. The saturating effect of diversity on log-scale community productivity would therefore suggest that the mechanisms by which fish communities are producing biomass may differ between temperate and tropical regions.

### Mechanistic patterns of community productivity
The decomposition of community-level biomass production may provide insight into the geographic discrepancies in BEF effects. Specifically, community productivity is the product of species total abundances and per-capita biomass production[48], which can be independently evaluated against abundance-emphasised diversity. While in temperate regions, increasing diversity positively correlated with increasing abundance (1.42 [1.41 to 1.43; Fig. 3a), this increase was much steeper in tropical regions (1.56 [1.55 to 1.58; Fig. 3b). Likewise, the observed decrease in per-capita productivity with diversity was

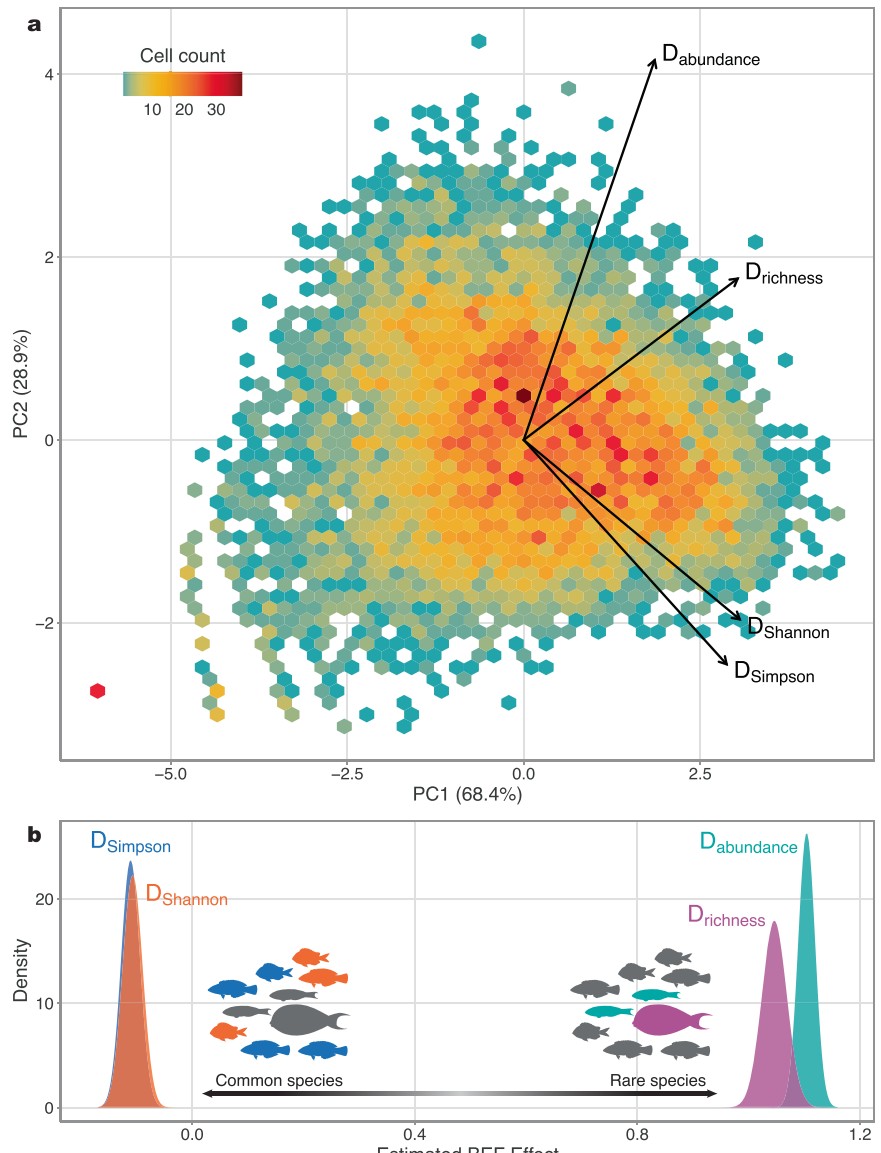

**Fig. 1 | Variation in Hill diversity metrics on ecosystem function. a** Principal component analysis (PCA) ordination displaying Hill diversity values (D) representing Simpson Index ($D_{Simpson}$), Shannon Index ($D_{Shannon}$), richness effects ($D_{richness}$), and abundance effects ($D_{abundance}$). Principal component axes 1 and 2 (PC1 and PC2, respectively) explain 68.4% and 28.9% of the variation of the data, respectively. Size of the arrows denote the magnitude and the direction denote the sign of each eigenvector. Hexagons are coloured based on the number of observations in a given bin. Note, the outlying red point towards negative PC1 and PC2 values is depicting the surveys that recorded a single species. **b** Posterior distributions of estimated biodiversity-ecosystem function (BEF) effects on biomass production (g 500 m⁻² day⁻¹) using Simpson Index (blue), Shannon Index (orange), richness effects (purple), and abundance effects (green) as quantifications of biodiversity. Both Simpson and Shannon indices emphasise common species, whereas richness and abundance effects emphasise rare species. All diversity indices were logged prior to analysis. Source data are provided as a Source Data file.

much steeper in the tropics (−0.41 [−0.44 to −0.38]; Fig. 3d) than in temperate regions (−0.33 [−0.35 to −0.31]; Fig. 3c). Therefore, the shift in configuration of the relationship between abundance and per-capita productivity (i.e., high per-capita productivity and low abundances towards low per-capita productivity and high absolute abundances) is strongest in the tropics, which suggests that the ecological mechanisms underpinning the BEF across marine fish communities (i.e., changes in per-capita biomass production versus abundance) are likely physiologically constrained in warm-water environments[15,50].

All Hill diversity indices had maximum values that were higher in the tropics (Fig. S3) and the largest concentration of high-diversity communities was above ~23 °C (Fig. 4). In warm, high-diversity systems, species are likely metabolically limited in their capacities to attain larger sizes[51,52]. Scaling temperature using the Boltzmann's-

Arrhenius relationship, a procedure accounting for non-linearities in how metabolic rates vary with temperature[53], changed the units, but not the shape of the relationship (see Methods; Table S2). Indeed, conspecifics found across temperature gradients tend to grow faster to smaller body sizes in warmer waters[54–56]. Similarly, species abundances scale negatively with body size[51,57,58], whereby conspecifics occurring in higher densities tend to exhibit smaller body sizes[15]. Taken together, these two relationships imply challenges with producing biomass in warm, high-abundance communities, possibly owing to trade-offs occurring between the increased costs of growth[59] and extrinsic environmental and ecological factors[60], such as competition in resource-limiting contexts. Although energetic constraints on total community fluxes[61] can also be modified by external energy subsidies (e.g., via pelagic pathways[32,62]), shifts towards higher abundances and

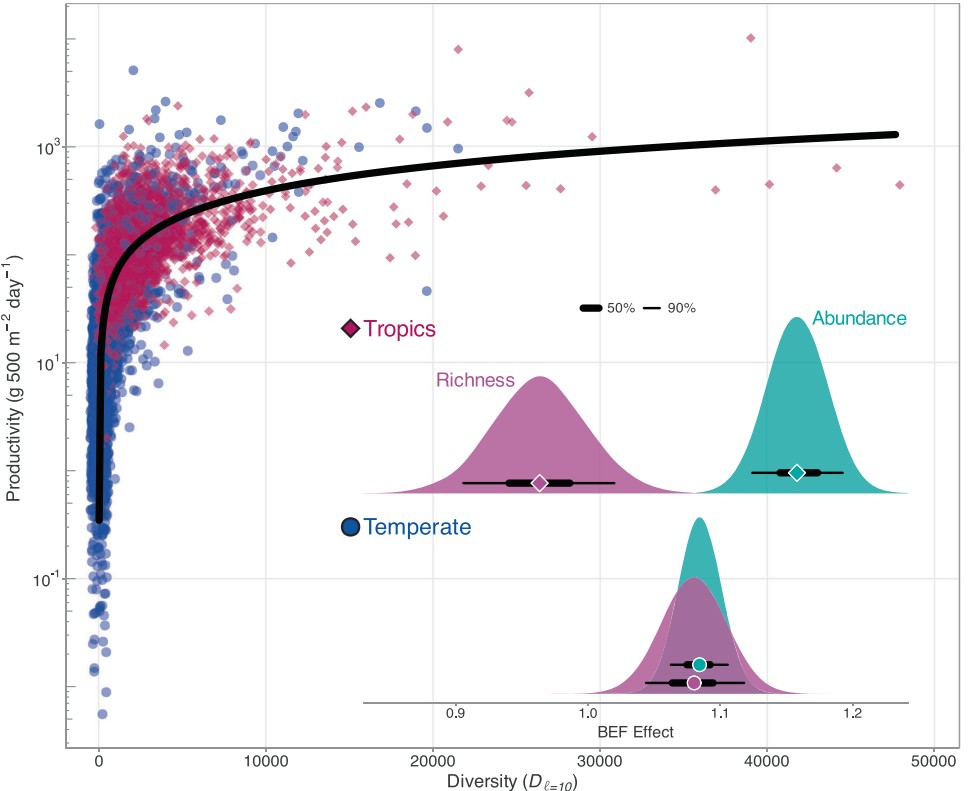

**Fig. 2 | Global biodiversity-ecosystem function relationship (BEF) across marine reef fishes.** Global BEF relationship across temperate (blue; circles) and tropical regions (red; diamonds). Each point is one survey and the thick black line is the median fitted trend from 400 randomly sampled draws. Note the y axis is on the log$_{10}$ scale and the points have been jittered (i.e., a very small random number has been added or removed) to improve clarity and interpretability of the figure. Inset: posterior distributions of modelled BEF relationships in tropical (top row) and temperate regions (bottom row) using Hill diversity emphasising abundance effects (green, $\ell = 10$) and richness effects (purple, $\ell = 1$). The points denote the median coefficient estimates, while the thick and thin bars denote the 50% and 90% credible intervals, respectively. Coefficient estimates can be found in Table S2. Source data are provided as a Source Data file.

lower per-capita productivity would also allow communities to circumvent physiological constraints due to warmer temperatures. Individuals from conspecific populations in warmer conditions may compensate for the enhanced difficulty in producing biomass via somatic growth by increasing their reproductive rates, thereby increasing their populations' relative abundances compared to their conspecifics in cold conditions. This involves trade-offs between size-based growth, mortality, and reproductive output (as suggested by the energetic-equivalence rule[63]). Thus, increased community productivity could emerge from such shifts. Indeed, declines in fishes' body sizes over time have been shown to be similarly matched by increases in intraspecific abundances, leading to relatively stable levels of community standing biomass[58]. While species interactions, such as intraspecific competition, can contribute to the trade-off between abundance and per-capita productivity (e.g., via self-thinning in resource-limited populations[64]), the physiological limits placed on individuals can likely scale to community-level biomass production, which could be imposed by environmental conditions working beyond resource availability.

The effects of diversity on abundance and per-capita productivity were dampened, but still distinguishable, across temperate regions, indicating that the energetic constraints placed on tropical communities may impact temperate communities as well, especially given that reef fishes tend to be larger towards higher latitudes[49,65]. However, given that species richness exhibited nearly an identical BEF effect on total community productivity as species abundances, these constraints are unlikely to be due to temperature alone and could be shaped by species interactions surrounding resource limitations. Temperate reef fish communities typically exhibit less community

compositional change[66] and higher evenness across latitudes[35], which could contribute to the relatively higher biomass production in these regions compared to the tropics (as denoted by their modelled intercepts, see Table S2). Increasing species richness in temperate regions had a greater impact on total community biomass production than tropical communities (Fig. 2), which mainly conforms to the traditional BEF paradigm: increasing richness correlates with increasing function delivery. In species-depauperate systems, such as those most often found in temperate areas, each additional species in a community would expand into relatively unoccupied ecological niches[3,35]. Consequently, with relatively low interspecific competition, each additional species could theoretically attain larger population sizes and, thus, deliver higher levels of functions, such as biomass production. Indeed, while evenness was not as strong of a predictor of functioning across geographic regions (see Fig. 1b), the dominance of high-performing species assessed within temperate regions has been shown to be a mechanistic driver of functioning in local marine communities (e.g., refs. 17–19). It is therefore possible that the strong, positive richness-emphasised BEF expressed in temperate regions across marine reef-associated fishes may reflect greater generalisations found throughout the BEF narrative. If the inherent biases derived from experimental BEF studies in high-latitude locations[67] are matched with the possibility that BEF effects can be stronger in temperate regions, then these biases could have been responsible for the disproportional emphasis on the dominant role of species richness, not abundances, on ecosystem functioning across observational BEF studies.

Given our empirical finding that high-diversity tropical communities are characterised by high abundances and low per-capita productivity, it appears that function delivery in hyperdiverse systems,

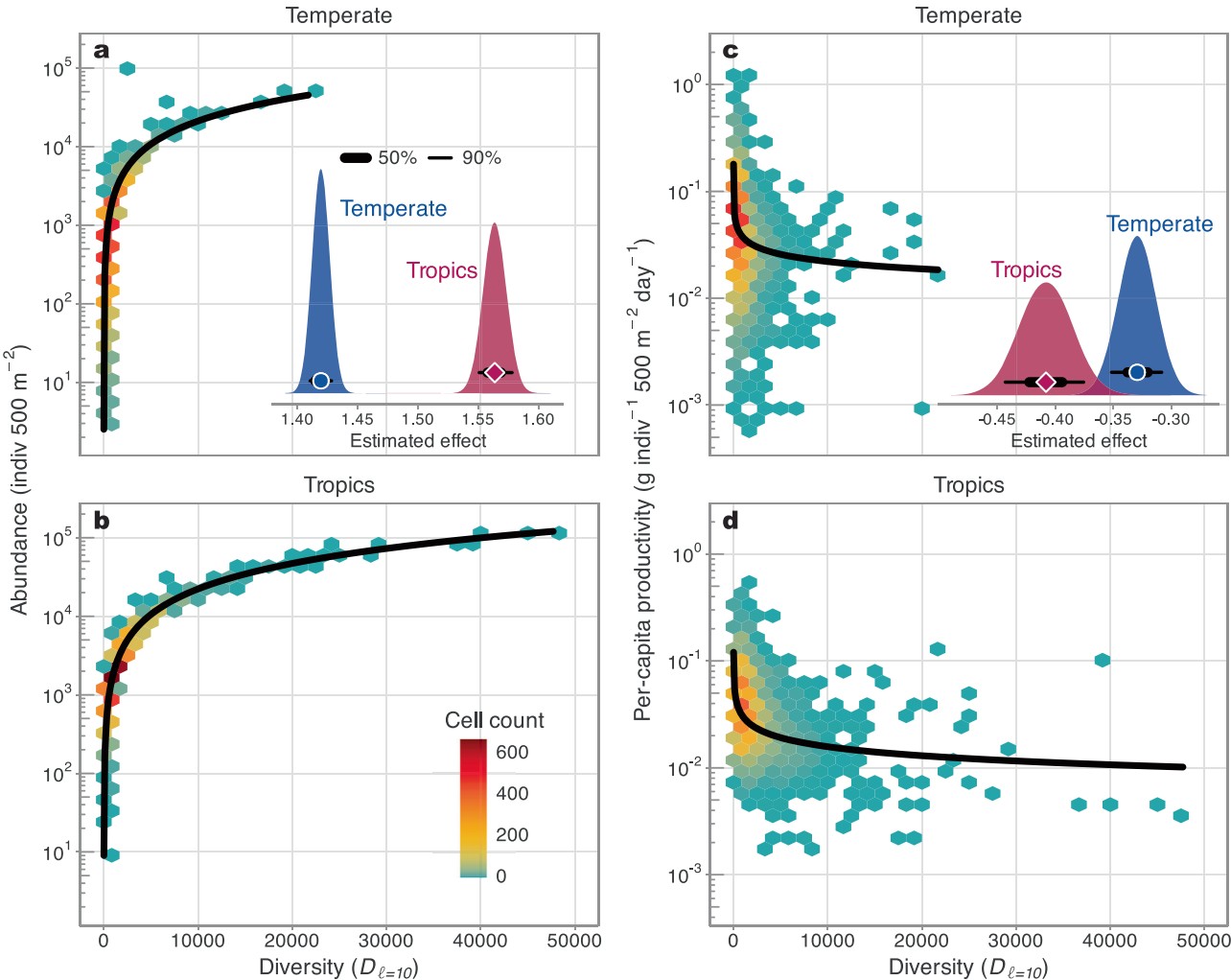

**Fig. 3 | Deconstructing the BEF across geographic regions.** The effects of Hill diversity emphasising abundance effects (i.e., diversity $D_{\ell=10}$) on total community abundance across **a** temperate and **b** tropical regions. The effects of Hill diversity on per-capita biomass production (g indiv$^{-1}$ 500 m$^{-2}$ day$^{-1}$) across **c** temperate and **d** tropical regions. Hexagons denote the density of the raw data and are coloured by the number of observations per bin. The thick black lines are the median fitted trends from 400 randomly sampled draws. The insets in (**a**, **c**) show the posterior distributions of the estimated slope coefficients for tropical and temperate regions in red and blue, respectively. The points denote the median estimates and the thick and thin bars represent the 50% and 90% credible intervals, respectively. Note the 50% credible intervals in (**a**) are nearly too narrow to be shown. Coefficient estimates for all models can be found in Table S2. Note all y axes are on the log$_{10}$ scale. Source data are provided as a Source Data file.

such as coral reefs, is primarily mediated by species' abundances, a factor which has largely been overlooked in previous BEF field studies. Indeed, high-diversity communities would require higher abundances to maintain steady increases in community-level biomass production as low-to-moderately high-diversity communities. In other words, increases in both species' abundances and biodiversity would need to be maintained to support continuously increasing levels of ecosystem functioning in high-diversity systems. Although the communities that are delivering some of the highest rates of function (in this case, biomass production) are necessarily the communities with the highest biodiversity, the difference in their observed capacity to provide this function relative to some communities with much lower biodiversity may be relatively small. Regardless of the relationship between diversity and biomass production, all tropical communities appear to be physiologically limited by temperature (Fig. 4; Fig. S4)[61]. Under future ocean warming scenarios, it is likely that increased temperature will further constrain the magnitude of function delivery expressed by communities that appear to be at their physiological limits, regardless of their species composition. The thermal tolerance-induced geographic ranges of cold-water fishes are continuously constrained by

ocean warming, indicating that cool-water species are likely to be more vulnerable than their tropical counterparts[68]. Therefore, BEF effects in temperate regions may shift towards tropical strategies, with increased emphases on abundances instead of species richness. We can therefore expect that with warming oceans, the temperature-related metabolic constraints on per-capita biomass production and the changes in ecological dynamics (e.g., species interactions) will reshape the productivity of fishes differently across geographic regions.

## Methods
We defined ecosystem functions as the movement or storage of energy or materials through an ecosystem[41,42]. We therefore adapted and applied a relatively novel approach for calculating biomass production[40], which is quantified as the rate of biomass produced via ontogenetic growth of all surveyed individuals over 500 m$^2$ over the course of one day, instead of using a proxy for functioning (e.g., standing stock biomass as seen in ref. 22). We first conducted a systematic literature review on all published von Bertalanffy growth models on marine teleost fishes from around the world to build a

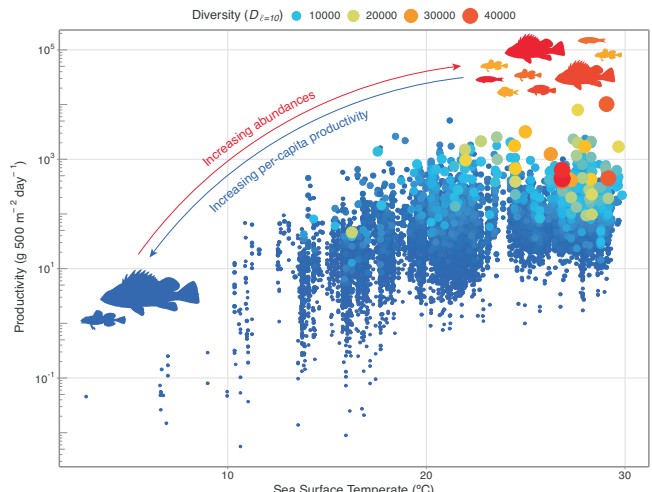

**Fig. 4 | Temperature limits the BEF.** Patterns in fish productivity (g 500 m$^{-2}$ day$^{-1}$) across sea surface temperature (°C) gradients. Points are sized and coloured based on their respective Hill diversity values emphasising abundance effects. Note the y axis is on the log$_{10}$ scale. Source data are provided as a Source Data file.

machine learning model to predict growth coefficients for fishes without empirically measured growth curves (as in ref. 39). Predictive models used a combination of ecological and environmental trait data, while also accounting for the aging method used (see below). We then used predicted growth coefficients to calculate the biomass production of reef fishes censused from one of the largest standardised global survey programs in the world: the Reef Life Survey program[37,38]. Although the Reef Life Survey program specifically censuses relatively shallow reef habitats, some of the species recorded have lower depth limits beyond 1000 m, hence we included all fishes in our systematic literature review. We then used Hill numbers to holistically capture multiple axes of biodiversity and assessed the BEF from the best-fitting Hill diversity metric. All models accounted for potential sources of environmental variation (i.e., water visibility, survey depth, temperature) and spatial heterogeneity (i.e., realm- and site-level effects).

## Data collation

We expanded on the framework developed by ref. 69 to generate standardised species-level estimates of somatic growth across marine teleost fishes. In short, we used a derivation of the von Bertalanffy Growth Model[70] known as $K_{max}$, which can be interpreted as the rate at which an individual along a specific growth trajectory would reach its asymptotic size if it grew to the species' maximum recorded size[69]. Please see ref. 69 for a detailed description of the quantification of $K_{max}$ and its associated equations. Because ref. 69 only considered coral reef fishes, we expanded on this by (1) first conducting a systematic literature review of all published Von Bertalanffy growth curves for all marine fishes, including reef and non-reef habitats; and (2) generating new predictive models to predict $K_{max}$ across all marine teleost fishes. Although we are estimating growth rates for fishes surveyed in relatively shallow coastal, hard substratum-associated habitats, fishes found in pelagic systems or from deeper waters have been recorded on Reef Life Survey transects. To ensure accurate growth estimates for all available fishes, we included observations from all non-reef habitats in our predictive models.

We collated a list of growth studies from FishBase[71] (accessed May 2022; $n = 5770$), which we supplemented with a systematic literature review (completed September 2022; $n = 1916$). We used the ISI Web of Science database using the search string *fish \* AND (growth OR von Bertalanffy)*, which resulted in 141649 articles. We filtered articles based on their titles and abstracts, removed all duplicates between the two sources, and corrected any typological/rounding errors recorded

from FishBase. We only included studies where species were collected from marine habitats (we omitted all freshwater species that were recorded in euryhaline habitats) and those where the discrete geographic locations were provided (see ref. 69 for a detailed explanation). This yielded 7686 growth curves encompassing 1480 species. From each study, we extracted the von Bertalanffy growth parameters $L_\infty$ and $K$, the length measurement type (i.e., total length, standard length, or fork length), the method used for aging (i.e., mark recapture, growth rings (e.g., otolith, scale, or vertebrae), length frequency, or unknown), and the geographic location. We then used the length-length conversion factors from FishBase to transform all estimates of $L_\infty$ to total length in cm.

With the mined growth values from the literature, we generated predictive Extreme Gradient Boosting models to use a series of traits and environmental variables to (1) explain variation in $K_{max}$ across fishes with empirically measured growth values and (2) predict $K_{max}$ for fishes lacking data (as in ref. 39). Specifically, we predicted growth trajectories for all the fishes recorded using a standardised global underwater visual census program, the Reef Life Survey program (accessed 16 December 2021). In short, reef fish communities were censused along 50 m-long transects along coral and rocky reefs around the world. Trained divers recorded the identity, size class, and abundance of all fishes across one-to-two 250 m$^2$-area blocks along a transect along relatively shallow depth ranges (<20 m). Detailed explanations of the survey methods can be found online (www. reeflifesurvey.com). We removed all surveys that were only composed of a single block from analyses ($n = 171$). For each species, including those from the mined literature and those recorded from the Reef Life Survey, we recorded the following trait data: maximum body size (total length, cm), trophic level (continuous), maximum depth (m), and position in the water column (categorical). Most of the trait data came from FishBase or the primary literature; when data were not available at the species level, we used estimates at the genus or family level. We used a 2° buffer around their respective locations and the years that they were sampled to generate an estimate of sea surface temperature (°C) using raster data from the National Oceanic and Atmospheric Administration (https://psl.noaa.gov/data/gridded/data.noaa.oisst. v2.html).

## Predicting growth

We combined the trait data (i.e., maximum body size, maximum depth, trophic level, and position in the water column) with estimates of sea surface temperature and the aging method (i.e., mark recapture, otolith rings, scale rings, other rings, length frequency, and unknown) in an Extreme Gradient Boosting framework[72,73] to explain the variation and predict the growth trajectories ($K_{max}$) of all the fishes recorded on Reef Life Survey (i.e., all surveyed fishes were given a predicted growth trajectory). Extreme Gradient Boosting models are a form of machine learning algorithm that combine multiple decision trees with a boosting algorithm, exhibit relatively high predictive accuracy, and are able to handle non-linearities and complex interactions[72,73]. We modelled $K_{max}$ using a Gamma loss function and selected hyperparameters using the two-step tuning method from refs. 39,69, which involved varying the learning rate (eta; 0.1–0.9), regularising parameter (gamma; 0.1–0.9), maximum tree depth (max_depth; 5, 10, or 15), and the subsample rate (0.1–0.9). The final hyperparameters reduced the negative log likelihood from 2.88 to 1.70 using the following hyperparameters: eta = 0.098, gamma = 0.89, max_depth = 10, subsample = 0.096.

We used a cross-validation procedure to model and predict $K_{max}$. The model was built and trained on 80% of the data and assessed against the remaining 20% test set. We assessed the model's bias by subtracting the predicted $K_{max}$ value from the test set against the observed value; a well-fitted model should have a bias close to zero. The model performance was also assessed by extracting the $R^2$ (i.e., the

goodness of fit) value from fitting a linear model between log(predicted) against log(observed) values from the test set. Finally, we predicted $K_{max}$ values for all the species recorded from the Reef Life Survey program. Due to the stochastic model-building procedure of Extreme Gradient Boosting models, we bootstrapped the entire process for 1000 iterations, using the `XGBoost` package v.1.4.1.1[72,73] in R v.4.1.0[74]. We chose to predict growth using the aforementioned method because our XGBoost models achieved a low median prediction bias of −0.005 (minimum, maximum: −0.01, 0.0008) and a high median precision $R^2$ of 0.72 (0.65, 0.76; Fig. S5), whereas phylogenetic predictive models (e.g., ref. [75]) can produce predictions with an accuracy as low as 51%. As in refs. [39],[69], maximum body size was the best predictor of $K_{max}$ (median variable importance: 43.1%), followed by sea surface temperature (21.1%), maximum depth (14.6%), trophic level (12.1%), aging method (5.3%), and position in the water column (3.7%; Fig. S6; Table S3).

## Calculating productivity

We calculated productivity following the methods set forth by ref. [40] using our newly estimated values of $K_{max}$. In short, productivity is measured as the amount of biomass acquired via somatic growth of all individuals in a community over the course of one day.

$$Productivity = Expected\ biomass_t - Observed\ biomass_{t-1} \quad (1)$$

Here, the expected biomass is the standing biomass plus the expected biomass arising via somatic growth, the observed biomass is the community standing biomass, and $t$ is the day. This generates a functional, process-based quantification of biomass production[41,42,76], which captures the underlying energetic and elemental fluxes experienced by the community compared to static proxies, such as standing stock biomass[25,40]. To place each individual within their growth trajectories, we randomly drew $K_{max}$ values from a truncated normal distribution from their 90% predicted quantile range. We then simulated the growth that would be expressed by each individual over the course of one day and used the difference between the two biomass estimates as the estimate of productivity. The total productivity was therefore calculated as the sum of the somatic growth of all individuals over the course of one day over 1000 bootstrapped simulations[40].

## Quantifying biodiversity

To capture multiple axes of biodiversity, namely abundance and richness, we used Hill numbers with different scaling parameters. Common indices used to quantify biodiversity (e.g., species richness) can be sensitive to rare species occurring in low abundances, whereas Hill numbers provide a continuous framework to estimate the effective number of species present[46,47]. Following ref. [12,48], we calculated Hill diversity (D) as:

$$D = \left( \sum_{i=1}^{S} p_i \left( \frac{1}{p_i} \right)^l \right)^{(1/l)} \quad (2)$$

Here, $p_1$, $p_2$, ..., $p_S$ are species' relative abundances for species richness $S$. This formulation of D allows for the differentiation between weights via abundance and rarity by controlling values of the scaling parameter $\ell$[12,48]. Specifically, setting $\ell$ to different values (or calculating the limit as $\ell$ approaches zero) generates different diversity indices and can drastically influence BEF relationships[48]. We explicitly tested the inverse Simpson Index ($\ell = -1$), exponentiated Shannon entropy ($\ell = 0$), species richness ($\ell = 1$), and the maximum abundance-emphasised value tested by ref. [48] ($\ell = 10$). Increasing emphasis on rarity will mathematically emphasise abundances: because the lowest possible count for a species is one individual (i.e., a singleton),

increasing rarity can only be achieved by increasing the total abundance of all other species in the community[48]. Therefore, Hill numbers that emphasise rarity more than species richness place de facto emphasis on communities with high total abundances[48]. We ran individual BEF analyses with each metric of biodiversity (i.e., Simpson Index, exponentiated Shannon entropy, species richness, and abundance index) and compared models by assessing relative effect sizes and using leave-one-out information criterion (LOOIC).

## Analyses

We first conducted a Principal Component Analysis (PCA) to assess the correlation of all Hill diversity metrics, which were log-transformed prior to all analyses to reduce the leverage of transects with disproportionally high diversity values. We presented figures with non-logged values, however, to highlight our hypotheses that each additional species unit would have the greatest impact in low-diversity systems, with the effect of each additional species unit having a saturating effect in log space. To assess the biodiversity-ecosystem function relationships, we used Bayesian generalised linear mixed-effects models (GLMMs) using productivity as the response variable. We individually modelled a global BEF model with productivity as a function of each Hill diversity metric (i.e., Simpson index, exponentiated Shannon entropy, species richness, and abundance index). We chose the top model as that which produced the smallest LOOIC value. All models included an interaction term between Hill diversity and latitudinal position; latitudinal position was separated into temperate and tropical locations based on their respective geographic realms. We specified a gamma error distribution for all models by specifying gamma shape ($\phi$) and rate ($\lambda_i$) parameters.

$$Gamma(y_i | \phi, \lambda_i) \quad (3)$$

$$\lambda_i = \frac{\phi}{\mu_i} \quad (4)$$

$$\log(\mu_i) = \beta_0 + \beta_D \log(x_D) + \beta_{trop} x_{trop} + \beta_{D*trop} \log(x_D) x_{trop} + \beta_{sst,vis,depth} \mathbf{X}_i + \alpha_{site} \quad (5)$$

$$\alpha_{site} \sim \alpha_{realm} + \varepsilon_{site} \quad (6)$$

$$\varepsilon_{site} \sim Normal(0, \sigma_{site}) \quad (7)$$

$$\beta_0 \sim Normal(0, 5) \quad (8)$$

$$\beta_{D, trop, D*trop, sst, vis, depth} \sim Normal(0, 1) \quad (9)$$

Here, $y_i$ is the observed community productivity for transect $i$, $\beta_O$ is the overall intercept, $\beta_{D*trop}$ is the interaction term between diversity and the tropics, $\beta_{D,trop,sst,vis,depth}$ are the estimated effects for Hill diversity (D), tropics (trop), sea surface temperate (sst), visibility (vis), and depth, respectively. We ran an additional model following the top model structure, but used the Boltzmann-Arrhenius relationship to convert temperature to inverse temperature (1/temperature x Boltzmann's constant) with temperature in Kelvin[53]. The term $\mathbf{X}_i$ is the design matrix of covariates. Site- and realm-level grouping factors are denoted as $\alpha_{site}$ and $\alpha_{realm}$, respectively, following a nested structure of site within realm. We specified weakly informative priors on all estimated effects. We modelled per-capita productivity following the same model structure as total community productivity, but we added log-transformed abundance as an offset in the model. Adding an offset in the model allows us to measure productivity on a per-capita basis, while still following the same underlying gamma error distribution.

We followed a similar model structure when modelling changes in abundance ($y_i$) with diversity between temperate and tropical regions, but we specified a negative binomial distribution using the inverse shape parameter ($\omega$) for overdispersion.

$$\text{Negative Binomial}(y_i \mid \mu_i, \omega) \tag{10}$$

$$\log(\mu_i) = \beta_0 + \beta_D \log(x_D) + \beta_{trop} x_{trop} + \beta_{D*trop} \log(x_D) x_{trop} + \beta_{sst, vis, depth} \mathbf{X}_i + \alpha_{site} \tag{11}$$

$$\text{Var}(y_i) = \mu_i + \frac{\mu_i^2}{\omega} \tag{12}$$

$$\alpha_{site} \sim \alpha_{realm} + \varepsilon_{site} \tag{13}$$

$$\varepsilon_{site} \sim \text{Normal}(0, \sigma_{site}) \tag{14}$$

$$\beta_0 \sim \text{Normal}(0, 5) \tag{15}$$

$$\beta_{D, trop, sst, vis, depth} \sim \text{Normal}(0, 1) \tag{16}$$

For all models, we scaled and centred all continuous fixed effects by subtracting the mean and dividing by the standard deviation prior to analysis. All models used four Markov-Chain Monte Carlo chains for 4000 iterations after an initial warmup phase of 2000 iterations using the `brms` package v.2.16.4[77]. We assessed model fit using posterior predictive checks and simulated residuals from the `DHARMa` package v.0.4.5[78] and achieved chain convergence for all estimate parameters (scale reduction factor Rhat <1.01) and effective sample sizes were all greater than 1800 (Table S2). Using Moran's I, we detected no spatial autocorrelation in the simulated residuals of any of the models. All modelled coefficients from GLMMs can be found in Table S2. No statistical method was used to predetermine sample size. Surveys comprising a single block were excluded (see above). There were no experiments involved, therefore blinding and randomisation were not used.

**Reporting summary**
Further information on research design is available in the Nature Portfolio Reporting Summary linked to this article.

## Data availability
All data were sourced from publicly available sources, including the peer-reviewed literature, FishBase (https://www.fishbase.org), and the Reef Life Survey program (https://reeflifesurvey.com). Data generated from these sources and used for analyses is available at Figshare (https://doi.org/10.6084/m9.figshare.26156344)[79]. Source data are provided as a Source Data file. Source data are provided with this paper.

## Code availability
The R (v.4.1.0) code used to run analyses and generate figures is available at Figshare (https://doi.org/10.6084/m9.figshare.26156344)[79].

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

## Acknowledgements

We thank the Reef Life Survey Team and all their volunteers for collecting and supplying the data, which was managed and sourced through Australia's Integrated Marine Observing System (IMOS). We also thank Simon J. Brandl, Robert P. Streit, Alexandre C. Siqueira, Sterling B. Tebbett, James Gahan, and Juliano Morais for valuable discussions; Hannah V. Watkins for statistical support; the Australian Research Council (ARC; DRB: FL190100062) and an ARC Laureate PhD scholarship and a Natural Sciences and Engineering Research Council of Canada Postgraduate Doctoral Scholarship (HFY) for financial support.

## Author contributions

Helen F. Yan: Conceptualisation, Formal analysis, Methodology, Data curation, Investigation, Visualisation, Writing – original draft. Renato A. Morais: Conceptualisation, Methodology, Writing – review & editing. David R. Bellwood: Conceptualisation, Writing – review & editing, Supervision, Funding acquisition.

## Competing interests

The authors declare no competing interests.
