## [Transparent Peer Review file · Nature Communications]

Species abundances surpass richness effects in the biodiversity-ecosystem function relationship across marine fishes

Corresponding Author: Ms Helen Yan

Version 0:

Reviewer comments:

Reviewer #1

(Remarks to the Author)

I have read and reviewed the manuscript "Species abundances surpass richness effects in global biodiversity-ecosystem function relationships" for Nature Communications. Overall, I think the study is very insightful, but somewhat lacks a few interpretations, especially in relation to temperate ecosystems. The most noteworthy result is how the authors disentangled the role of abundance-based biodiversity indices on biomass production and found that abundance metrics do not influence the diversity-function relationships similarly between the tropics and temperate reef fish communities (Figures 2 and 3). I believe my comments can be addressed in a revision, and I hope they will help the authors finalize their work.

L24-26: better not to overstate what was done by precisizing this is not about all fish, and it's not truly global.

L51-53: while I agree abundance got a lot less attention in the literature overall, there are many papers that have investigated it, especially in the marine environment covering phytoplankton, fish, macrophytes, and the deep sea. I think it is important to give a better representation of that literature that has developed in the last 10-20 years. Instead of expanding directly to birds and plants, I would have appreciated to see more marine papers covering abundance effects. There's many more than the ones I mention below, especially from fisheries systems.

- Dangles, Olivier, and Björn Malmqvist. 2004. "Species Richness–Decomposition Relationships Depend on Species Dominance." *Ecology Letters* 7 (5): 395–402. <https://doi.org/10.1111/j.1461-0248.2004.00591.x>.
- Engelhardt, Katharina A. M., and Mark E. Ritchie. 2001. "Effects of Macrophyte Species Richness on Wetland Ecosystem Functioning and Services." *Nature* 411 (6838): 687–89. <https://doi.org/10.1038/35079573>.
- Hillebrand, Helmut, Danuta M. Bennett, and Marc W. Cadotte. 2008. "Consequences of Dominance: A Review of Evenness Effects on Local and Regional Ecosystem Processes." *Ecology* 89 (6): 1510–20. <https://doi.org/10.1890/07-1053.1>.
- Hodapp, Dorothee, Sandra Meier, Friso Muijsers, Thomas H. Badewien, and Helmut Hillebrand. 2015. "Structural Equation Modeling Approach to the Diversity-Productivity Relationship of Wadden Sea Phytoplankton." *Marine Ecology Progress Series* 523 (March):31–40. <https://doi.org/10.3354/meps11153>.
- Lehtinen, Sirpa, Timo Tamminen, Robert Ptacnik, and Tom Andersen. 2017. "Phytoplankton Species Richness, Evenness, and Production in Relation to Nutrient Availability and Imbalance." *Limnology and Oceanography* 62 (4): 1393–1408. <https://doi.org/10.1002/lno.10506>.
- Maureaud, Aureore, Ken H. Andersen, Lai Zhang, and Martin Lindegren. 2020. "Trait-Based Food Web Model Reveals the Underlying Mechanisms of Biodiversity–Ecosystem Functioning Relationships." *Journal of Animal Ecology* 89 (6): 1497–1510. <https://doi.org/10.1111/1365-2656.13207>.
- Maureaud, Aureore, Dorothee Hodapp, P. Daniël van Denderen, Helmut Hillebrand, Henrik Gislason, Tim Spaanheden Dencker, Esther Beukhof, and Martin Lindegren. 2019. "Biodiversity–Ecosystem Functioning Relationships in Fish Communities: Biomass Is Related to Evenness and the Environment, Not to Species Richness." *Proceedings of the Royal Society B: Biological Sciences* 286 (1906): 20191189. <https://doi.org/10.1098/rspb.2019.1189>.

L53-61: This paragraph needs revision based on the literature mentioned above.

L74-76: Looking at Figure S1, yes, the RLS program covers many places in the oceans, but it would be useful to precise

which habitats, and its sampling sites are mostly concentrated around the tropics, which makes it an interesting dataset since most marine community datasets are data-poor in the tropics. This is because of the fish communities sampled (reef fishes mostly). I think the description could be more accurate here.

Figure 1b: I don't understand why there is not a y axis displayed.

L126-131: I don't understand what this result refers to.

L201-217: I think this interpretation should be balanced thinking of the following paper, given the data sources between the two papers on growth rates are similar:

- Denderen, Daniël van, Henrik Gislason, Joost van den Heuvel, and Ken H. Andersen. 2020. "Global Analysis of Fish Growth Rates Shows Weaker Responses to Temperature than Metabolic Predictions." *Global Ecology and Biogeography* 29 (12): 2203–13. <https://doi.org/10.1111/geb.13189>.

L218-238: What kind of literature are you using to compare it with? The conclusions of this paper could only be true for reef fish communities in temperate areas, and not all temperate fish communities.

L227-231: this may be true in your study, but not across other works on temperate marine communities using observational data (see literature mentioned above). Balancing this result with other literature in the field would give better context.

L235-239: agreed that there is a strong difference in availability of datasets across the globe for BEF research and in general, as explained in the Clarke et al., paper from 2017. However, these are for experiments, not observational studies, and it is not clear to me the same bias applies in the marine environment in general. There are not that many BEF marine papers in the oceans on fish communities, whether temperate or tropical. Actually, I think the RLS has been one of the first big marine fish datasets to be tested. It would be great if the authors would contextualize their study in the field of BEF studies in the oceans and fish specifically. Given that, is it true that richness effects have been more emphasized in BEF research because temperate marine fish communities are more studied? My expectation is the opposite, that richness effects may not matter as much, and that tropical fish communities have been more studied for BEF relationships than temperate fish communities with observational datasets.

L256-259: agreed, though this to me is not a novel conclusion. How about how climate change will impact temperate versus tropical communities based on the BEF findings of the study? This would be more aligned with the overall findings of the papers and how to take them further. Taking L229-231 further, what can be said about temperate ecosystems warming up in terms of BEF relationships?

L323-325: better trait inference methods now exist than using the genus and family level, such as the following paper already cited later in the manuscript:

- Thorson, James T., Aurore A. Maureaud, Romain Frelat, Bastien Mériqot, Jennifer S. Bigman, Sarah T. Friedman, Maria Lourdes D. Palomares, Malin L. Pinsky, Samantha A. Price, and Peter Wainwright. 2023. "Identifying Direct and Indirect Associations among Traits by Merging Phylogenetic Comparative Methods and Structural Equation Models." *Methods in Ecology and Evolution* 14 (5): 1259–75. <https://doi.org/10.1111/2041-210X.14076>.

L352-356: this is interesting that the authors used genus and family trait inference manually but that they didn't use phylogenetic predictive models. How many traits were inferred with genus and family levels and how much would that influence the overall performance of the boosted regression trees in estimating the growth rates?

I did not get to see the code underlying the study and this to me is disappointing, as it would improve transparency and reproducibility of the results. I strongly encourage the authors to make their code open for review and for the readers to reproduce the results or re-use the methods for other studies alongside the dataset.

Reviewer #2

(Remarks to the Author)

General Comments:

I think this is an interesting, important but rather dense paper. The authors used a biodiversity-ecosystem function (BEF) approach to evaluate fish communities globally and found distinct differences between temperate and tropical marine ecosystems. The study is comprehensive and reaches the important conclusion that at higher sea temperatures fish abundances increase while per-capita productivity of fish declines. While the authors drew this conclusion based on thousands of studies, I worry that an audience of non-specialists may find the approach and conclusions a bit opaque.

There is both a long history and numerous ways to express the diversity of species. The authors used "Hill diversity" that weights abundance over other indices such as species richness. This is fine but a clear explanation of why this approach was used and if it could skew conclusions would be helpful for the general readers of this journal.

The study's overarching contribution is in its global scope. I think there is terrific value in developing geographic baselines against which anthropogenic change can be gauged. However, generalizations at this scale often can die from 10,000 paper cuts. Skeptics may simply join a chorus of "buts". That said, I'll introduce my "but" to which I hope the authors can respond.

Fig. 4 shows a striking global pattern of increasing sea surface temperature and productivity. This is said to result in increasing abundances in warmer regimes and increasing per-capita productivity in cooler regimes. There is no mention of why productivity is higher at warmer temperatures. However, average daylength and annual solar intensity is highest in the tropics and this undoubtedly affects annual primary productivity. Line 406 stated: " All models included an interaction term between Hill diversity and latitudinal position; latitudinal position was separated into temperate and tropical locations based on their respective geographic realms." So, why is sea temperature rather than latitude or annual solar radiation the independent variable in Fig. 4?

I like to see big picture studies like this published. It also gives us a better sense of deviations in recent decades due to human alterations of our biosphere. I hope the authors can address my concerns.

Specific comments:

Line 20: "Biodiversity" meaning should be clarified. The term "biodiversity" is a contraction of the phrase "biological diversity" (Wilson and Peter 1988). It was intended to encompass all scales of diversity from genomic to species, populations, communities, ecosystems, and landscapes. The term has been used in some studies as synonymous with species diversity. I know that is not the intent of the authors but a brief clarification here might be useful for readers of this journal.

Line 34: What is included in "ecosystem functioning"? Ecosystem stability? i.e. resistance of fundamental change in structure and function?

Line 72 states: "Here, productivity is defined as the instantaneous biomass accumulated via ontogenetic growth of all individuals (without mortality) over the survey area over the course of one day". Productivity is always a rate so the word "instantaneous" is confusing and unnecessary.

Line 82: "Hill diversity" is so important to this paper that a reference is warranted.

Line 102: If Hill diversity that emphasizes abundance over species richness is this study's operational definition of diversity, doesn't it skew results towards the title of the study: "Species abundances surpass richness effects in global biodiversity-ecosystem function relationships"

Line 107: Give reference for Hill 1973 here.

Line 135: Some colorblind people cannot differentiate these colors.

Line 137: What does "jittered" mean?

Line 153: " species abundances may be the primary underlying driver of productivity across tropical fish communities". This sounds like circular reasoning. You define productivity as mass per time so abundance must be the result not the driver of productivity (note it is the dependent variable in Fig. 4). Why isn't sunlight and low annual temperature variations the drivers of primary and secondary productivity?

Line 176: Won't changes in per-capita biomass be spatially constrained in warmer water due to increased respiration?

Line 194: " In warm, high-diversity systems, species are likely metabolically limited in their capacities to attain larger sizes." That is Pauly's GOLT (gill oxygen limitation theory) and it should be cited here.

Reviewer #3

(Remarks to the Author)

The manuscript entitled "Species abundances surpass richness effects in global biodiversity-ecosystem function relationships" explored the relationship between biodiversity and ecosystem function (BEF) across over 1000 marine fish species globally. The authors found that species abundance effects were more influential than species richness effects in global BEF relationship. This study applies several statistical models to multiple datasets from literature review, FishBase, and field surveys. The authors did a great job on organizing the analyses and presenting the results. This is a very interesting study, and it could be attractive to a lot of fishery scientists and managers.

This manuscript is well-written; however, the Methods section may need to be a little more detailed. I have a few concerns about the data collection and statistical models in this study for the authors to consider and/or clarify.

1. The authors mentioned that the fish trait data came from multiple sources (e.g., FishBase or literature review), and they also pointed out that they corrected some recorded errors from FishBase, which is a little bit confusing. If the authors could clearly tell readers how many species were explored from literature review and how many from FishBase, respectively, it will be helpful for readers to better understand the data sources. In addition, it is even better to know what kind of or how many errors exist in FishBase, which will save a lot of effort for the researchers who are using FishBase as well.

2. The models used to predict K_{max} are Extreme Gradient Boosting (XGBoost) models. However, the authors used the term "boosted regression trees (BRT)" in the text. XGBoost and BRT are quite different in model fitting and hyperparameter setting. Here I feel "XGBoost" is a better term to use.

3. The authors did not provide the von Bertalanffy growth equation, but instead, they directly talked about using XGBoost models to predict growth, which seems a little confusing. I feel it would be better if the authors could start with writing out the

von Bertalanffy equation and then derive the equations from it to estimate Kmax.

4. In the XGBoost model hyperparameters, the value of subsample is way too small (<0.1). Since there is no corresponding R code attached in this manuscript, I cannot help testing it accordingly. I hope the author may double-check the value of it.
 5. In the Kmax prediction model validation paragraph, it is great for the authors calculating model's bias. It would be better to calculate the estimator's precision as well. The accuracy would be the combination of bias and precision.
 6. In the Calculating productivity paragraph, please add and then explain the formula to calculate productivity.
- Some minor issues are listed below.

Line 53: Please move number "17" into the parenthesis.

Line 101: Please explain leave-one-out information criterion (LOOIC).

Line 222: Please capitalize "f" in "BEf".

Line 311: Both "regression tree" and "regression trees" were used in the text, please keep them consistent.

Line 348: Please explain "r2". It was written "R2" in Figure S4. Please be consistent.

Line 352: The sentence "We did not..." is not clear. Please try to reword it.

Line 445: Please explain "Rhat".

Line 446: How to detect spatial autocorrelation? Not clear.

Version 1:

Reviewer comments:

Reviewer #2

(Remarks to the Author)

The revised manuscript and detailed responses are all fine with me. I remain a bit concerned with how far latitudinal trends to reef fish apply.

Reviewer #3

(Remarks to the Author)

The authors have responded all my comments and fully addressed my concerns. Good job!

Reviewer #1

I have read and reviewed the manuscript “Species abundances surpass richness effects in global biodiversity-ecosystem function relationships” for Nature Communications. Overall, I think the study is very insightful, but somewhat lacks a few interpretations, especially in relation to temperate ecosystems. The most noteworthy result is how the authors disentangled the role of abundance-based biodiversity indices on biomass production and found that abundance metrics do not influence the diversity-function relationships similarly between the tropics and temperate reef fish communities (Figures 2 and 3). I believe my comments can be addressed in a revision, and I hope they will help the authors finalize their work.

RESPONSE: Thank you for your constructive and positive feedback on our manuscript. We greatly appreciate your suggestions and believe that they have strengthened our work. Please note that the line numbers pertain to the track-changed version of the manuscript.

L24-26: better not to overstate what was done by precisizing this is not about all fish, and it’s not truly global.

RESPONSE: We have removed “global” from this sentence (line 28).

L51-53: while I agree abundance got a lot less attention in the literature overall, there are many papers that have investigated it, especially in the marine environment covering phytoplankton, fish, macrophytes, and the deep sea. I think it is important to give a better representation of that literature that has developed in the last 10-20 years. Instead of expanding directly to birds and plants, I would have appreciated to see more marine papers covering abundance effects. There’s many more than the ones I mention below, especially from fisheries systems.

- Dangles, Olivier, and Björn Malmqvist. 2004. “Species Richness–Decomposition Relationships Depend on Species Dominance.” *Ecology Letters* 7 (5): 395–402. <https://doi.org/10.1111/j.1461-0248.2004.00591.x>.
- Engelhardt, Katharina A. M., and Mark E. Ritchie. 2001. “Effects of Macrophyte Species Richness on Wetland Ecosystem Functioning and Services.” *Nature* 411 (6838): 687–89. <https://doi.org/10.1038/35079573>.
- Hillebrand, Helmut, Danuta M. Bennett, and Marc W. Cadotte. 2008. “Consequences of Dominance: A Review of Evenness Effects on Local and Regional Ecosystem Processes.” *Ecology* 89 (6): 1510–20. <https://doi.org/10.1890/07-1053.1>.
- Hodapp, Dorothee, Sandra Meier, Friso Muijsers, Thomas H. Badewien, and Helmut Hillebrand. 2015. “Structural Equation Modeling Approach to the Diversity-Productivity Relationship of Wadden Sea Phytoplankton.” *Marine Ecology Progress Series* 523 (March):31–40. <https://doi.org/10.3354/meps11153>.
- Lehtinen, Sirpa, Timo Tamminen, Robert Ptacnik, and Tom Andersen. 2017. “Phytoplankton Species Richness, Evenness, and Production in Relation to Nutrient Availability and Imbalance.” *Limnology and Oceanography* 62 (4): 1393–1408. <https://doi.org/10.1002/lno.10506>.
- Maureaud, Aurore, Ken H. Andersen, Lai Zhang, and Martin Lindegren. 2020. “Trait-Based Food Web Model Reveals the Underlying Mechanisms of Biodiversity–Ecosystem

Functioning Relationships.” *Journal of Animal Ecology* 89 (6): 1497–1510.
<https://doi.org/10.1111/1365-2656.13207>.

- Maureaud, Aurore, Dorothee Hodapp, P. Daniël van Denderen, Helmut Hillebrand, Henrik Gislason, Tim Spaanheden Dencker, Esther Beukhof, and Martin Lindegren. 2019. “Biodiversity–Ecosystem Functioning Relationships in Fish Communities: Biomass Is Related to Evenness and the Environment, Not to Species Richness.” *Proceedings of the Royal Society B: Biological Sciences* 286 (1906): 20191189.
<https://doi.org/10.1098/rspb.2019.1189>.

RESPONSE: Thank you for providing these references, we have updated the examples to include more freshwater/marine examples.

L53-61: This paragraph needs revision based on the literature mentioned above.

RESPONSE: We have added the following sentence to this paragraph based on the previously provided resources (lines 60-62).

Dominance, which indirectly considers species relative abundances, has been used in observational BEF studies, however, it can have variable effects across both functions (e.g. ref²⁴) and ecosystems (e.g. ref^{6,17,25}).

L74-76: Looking at Figure S1, yes, the RLS program covers many places in the oceans, but it would be useful to precise which habitats, and its sampling sites are mostly concentrated around the tropics, which makes it an interesting dataset since most marine community datasets are data-poor in the tropics. This is because of the fish communities sampled (reef fishes mostly). I think the description could be more accurate here.

RESPONSE: We have reworded this sentence (lines 78-81). It now reads as:

By building predictive models from 7,686 growth curves across 1,480 species, we were able to estimate the local per-capita productivity of marine fishes from 2,957 sites comprising coral and rocky reefs, spanning tropical to polar locations (Fig. S1).

Figure 1b: I don’t understand why there is not a y axis displayed.

RESPONSE: We originally chose not to include an axis as it only displays density. We have now added a y axis to this plot.

L126-131: I don’t understand what this result refers to.

RESPONSE: We have removed this sentence to avoid confusion.

L201-217: I think this interpretation should be balanced thinking of the following paper, given the data sources between the two papers on growth rates are similar:

- Denderen, Daniël van, Henrik Gislason, Joost van den Heuvel, and Ken H. Andersen. 2020. “Global Analysis of Fish Growth Rates Shows Weaker Responses to Temperature than Metabolic

Predictions.” *Global Ecology and Biogeography* 29 (12): 2203–13.
<https://doi.org/10.1111/geb.13189>.

RESPONSE: This is a good paper to include in this paragraph, thank you for bringing it to our attention. Van Denderen et al. found that there was still a positive relationship, albeit weaker than expected based on theory, between temperature and growth across fishes. They stipulated that this weaker relationship could be due to different environmental and ecological dynamics, which we also highlight in this section. We have therefore rewritten the sentence on lines 214-217 to be:

Taken together, these two relationships imply challenges with producing biomass in warm, high-abundance communities, possibly owing to trade-offs occurring between the increased cost of growth⁵⁹ and extrinsic environmental and ecological factors⁶⁰, such as competition in resource-limiting contexts.

L218-238: What kind of literature are you using to compare it with? The conclusions of this paper could only be true for reef fish communities in temperate areas, and not all temperate fish communities.

RESPONSE: This paragraph mainly refers to literature on reef fish communities, we have therefore adjusted the text to clarify that our findings pertain primarily to reef-associated fishes (lines 239, 253).

L227-231: this may be true in your study, but not across other works on temperate marine communities using observational data (see literature mentioned above). Balancing this result with other literature in the field would give better context.

RESPONSE: The majority of previous research indicates that within temperate marine communities, the dominance of high-performing species outweighs the effects of species richness. We know, however, that across reef fishes, temperate communities generally display higher evenness than tropical communities (i.e. the opposite of dominance; see ref. 34). These inconsistencies between abundances and species richness are likely due to the relative scales that are assessed. Much of the literature (including those that were provided) are comparing temperate communities to one another, whereas our work here is making macroecological comparisons between temperate and tropical communities. Indeed, locally, dominant species may produce higher levels of functioning within temperate communities (e.g. references 16-18); but when assessed across macroecological scales, they still display higher evenness than tropical communities (see ref. 34). To avoid confusion with regards to within- versus across-region comparisons, we have added a sentence further along this paragraph to explain these differences in scale (lines 249-252):

Indeed, while evenness was not a strong predictor of functioning across geographic regions (see Fig. 1b), the dominance of high-performing species assessed within temperate regions has been shown to be a mechanistic driver of functioning in local marine communities (e.g. refs¹⁶⁻¹⁸).

L235-239: agreed that there is a strong difference in availability of datasets across the globe for BEF research and in general, as explained in the Clarke et al., paper from 2017. However, these are for experiments, not observational studies, and it is not clear to me the same bias applies in the marine environment in general. There are not that many BEF marine papers in the oceans on fish communities, whether temperate or tropical. Actually, I think the RLS has been one of the first big marine fish datasets to be tested. It would be great if the authors would contextualize their study in the field of BEF studies in the oceans and fish specifically. Given that, is it true that richness effects have been more emphasized in BEF research because temperate marine fish communities are more studied? My expectation is the opposite, that richness effects may not matter as much, and that tropical fish communities have been more studied for BEF relationships than temperate fish communities with observational datasets.

RESPONSE: We agree with the reviewer, there are not many BEF marine papers in the oceans on fish communities. Of the ones that do exist, those assessed across macroecological scales tend to use proxies of function (e.g. standing biomass) instead of actual functions (e.g. refs. 21 and 65). Indeed, the only other relatively global BEF analysis on marine fishes (ref. 21) also uses the RLS data, but instead quantifies standing biomass, which has been empirically decoupled from productivity (e.g. Morais et al., 2020). The majority of other studies are typically analysed within a region (e.g. ref. 18; Lefcheck et al., 2019). To tease apart the effect of these differences would require a systematic literature review, and even then, there may be challenges in describing geographic biases.

This is an interesting question, however, the main point that we are trying to convey in this section is to highlight that many people tend to extrapolate the findings, and associated biases, from BEF studies (especially those arising from experimental studies) to explain patterns observed in nature. To clarify this objective, we have modified this sentence (lines 254-258), which now reads as:

If the inherent biases derived from experimental BEF studies in high-latitude locations⁶⁶ are matched with the possibility that BEF effects can be stronger in temperate regions, then these biases could have been responsible for the disproportional emphasis on the dominant role of species richness, not abundances, on ecosystem functioning across observational BEF studies.

References

Lefcheck, J.S., Innes-Gold, A.A., Brandl, S.J., Steneck, R.S., Torres, R.E. and Rasher, D.B., 2019. Tropical fish diversity enhances coral reef functioning across multiple scales. *Science advances*, 5(3), p.eaav6420.

Morais, R.A., Connolly, S.R. and Bellwood, D.R., 2020. Human exploitation shapes productivity–biomass relationships on coral reefs. *Global Change Biology*, 26(3), pp.1295-1305.

L256-259: agreed, though this to me is not a novel conclusion. How about how climate change will impact temperate versus tropical communities based on the BEF findings of the study? This would be more aligned with the overall findings of the papers and how to take them further.

Taking L229-231 further, what can be said about temperate ecosystems warming up in terms of BEF relationships?

RESPONSE: To better address these issues, we have rephrased the final few sentences (lines 272-282) of this paragraph to read:

Under future ocean warming scenarios, it is likely that increased temperature will further constrain the magnitude of function delivery expressed by communities that appear to be at their physiological limits, regardless of their species composition. The thermal tolerance-induced geographic ranges of cold-water fishes is continuously constrained by ocean warming, indicating that cool-water species are likely to be more vulnerable than their tropical counterparts⁶⁷. Therefore, BEF effects in temperate regions may shift towards tropical strategies, with increased emphases on abundances instead of species richness. We can therefore expect that with warming oceans, the temperature-related metabolic constraints on per-capita biomass production and the changes in ecological dynamics (e.g. species interactions) will reshape the productivity of fishes differently across geographic regions.

L323-325: better trait inference methods now exist than using the genus and family level, such as the following paper already cited later in the manuscript:

- Thorson, James T., Aurore A. Maureaud, Romain Frelat, Bastien Mérigot, Jennifer S. Bigman, Sarah T. Friedman, Maria Lourdes D. Palomares, Malin L. Pinsky, Samantha A. Price, and Peter Wainwright. 2023. "Identifying Direct and Indirect Associations among Traits by Merging Phylogenetic Comparative Methods and Structural Equation Models." *Methods in Ecology and Evolution* 14 (5): 1259–75. <https://doi.org/10.1111/2041-210X.14076>.

RESPONSE: We agree that using genus- or family-level traits can be relatively coarse, however the work by Thorson et al., while helpful for categorical traits, is still limited in its capacity to quantify continuous traits. Indeed, their cross-validation assessments indicate that their models can explain as little as 51% of the variance of the validation set, which performed particularly poorly for measuring growth and trophic level. Additionally, their models are predicated on using structural equation models to generate covariance matrices between traits, yet many of the links shown in their Figure 2 have not been empirically tested in reef fishes, a problem that is further complicated by their inclusion of non-reef-associated fishes (e.g. freshwater fishes). Despite using genus/family-level traits, our models still had a higher predictive precision (R^2 range = 0.65-0.76) than Thorson et al., which is why we chose to retain our approach of using genus/family-level means.

L352-356: this is interesting that the authors used genus and family trait inference manually but that they didn't use phylogenetic predictive models. How many traits were inferred with genus and family levels and how much would that influence the overall performance of the boosted regression trees in estimating the growth rates?

RESPONSE: We were able to use species-level traits for 80-90% of the species recorded, with only 8-16% of the species using genus-level traits. The variables with the most influence in our Extreme Gradient Boosted models were maximum body size and sea

surface temperature, accounting for 43.1% and 21.1% of the total variable importance, respectively. While sea surface temperature estimates were all extracted from raster data, we were able to use species-level estimates of maximum body size for 89.5% of the species used herein, which accounts for more than 90% of the total biomass observations. This pattern was consistent across all traits, indicating that our use of genus- or family-level traits would likely have very little impact on the estimation of growth rates. Indeed, our models consistently achieved a low prediction bias and relatively high accuracy (lines 380-384).

I did not get to see the code underlying the study and this to me is disappointing, as it would improve transparency and reproducibility of the results. I strongly encourage the authors to make their code open for review and for the readers to reproduce the results or re-use the methods for other studies alongside the dataset.

RESPONSE: We apologise for this oversight during the review process. All code and data have been provided on Figshare.

<https://figshare.com/s/93dc995645c1fea3a9a2>

Reviewer #2:

General Comments:

I think this is an interesting, important but rather dense paper. The authors used a biodiversity-ecosystem function (BEF) approach to evaluate fish communities globally and found distinct differences between temperate and tropical marine ecosystems. The study is comprehensive and reaches the important conclusion that at higher sea temperatures fish abundances increase while per-capita productivity of fish declines. While the authors drew this conclusion based on thousands of studies, I worry that an audience of non-specialists may find the approach and conclusions a bit opaque.

RESPONSE: Thank you for your attention with our manuscript, we greatly appreciate your comments and assessment of our work. Please note the line numbers are in reference to the track-changed version of the manuscript.

There is both a long history and numerous ways to express the diversity of species. The authors used "Hill diversity" that weights abundance over other indices such as species richness. This is fine but a clear explanation of why this approach was used and if it could skew conclusions would be helpful for the general readers of this journal.

RESPONSE: We thank the reviewer for the comment as it highlights one of our oversights. To clarify the use of Hill diversities and how this has shaped our results, we have added clearer explanations where we introduce Hill diversity (lines 87-98). This reads as:

Here, we used Hill diversity^{45,46} as a means to quantify biodiversity and evaluate different diversity indices in explaining variation in biomass production. In short, Hill diversity is a general measure of species diversity, which can be scaled to calculate common diversity indices (e.g. Simpson Index, Shannon Entropy), but allows all measures to be expressed using the same units (i.e. units of species)¹². Unlike other diversity metrics, changes in Hill diversity values intuitively reflect changes in the community; for example, the loss of species in a community would proportionally scale with a decrease in Hill diversity¹². Because Hill diversity is a generalised equation, we can generate commonly used diversity indices within a single equation by varying the scaling parameter ℓ within a single equation (see methods)^{12,47}. By changing the scaling parameter ℓ , we can specifically emphasise common species (e.g. Simpson Index, Shannon entropy) or rare species (e.g. abundance effects) more than richness, thereby placing communities along an evenness-rarity continuum^{12,47}.

We have also clarified our results in the abstract to highlight the strength of using Hill numbers (lines 30-33).

The study's overarching contribution is in its global scope. I think there is terrific value in developing geographic baselines against which anthropogenic change can be gauged. However, generalizations at this scale often can die from 10,000 paper cuts. Skeptics may simply join a chorus of "buts". That said, I'll introduce my "but" to which I hope the authors can respond. Fig. 4 shows a striking global pattern of increasing sea surface temperature and productivity. This is said to result in increasing abundances in warmer regimes and increasing per-capita productivity in cooler regimes. There is no mention of why productivity is higher at warmer temperatures. However, average daylength and annual solar intensity is highest in the tropics and this undoubtedly affects annual primary productivity. Line 406 stated: " All models included an interaction term between Hill diversity and latitudinal position; latitudinal position was separated into temperate and tropical locations based on their respective geographic realms." So, why is sea temperature rather than latitude or annual solar radiation the independent variable in Fig. 4?

RESPONSE: This is a very good point, thank you for bringing it to our attention. We completely agree that there are factors other than temperature that covary with latitude, such as growing season and primary productivity, that could influence the productivity of fishes. We do believe, however, that temperature provides a stronger theoretical basis for describing patterns in productivity than other environmental variables (such as solar irradiance or chlorophyll-a concentration). The causal links between primary productivity and temperature relationships are complex and uncertain across many ecosystems (including reef fishes). For example, temperature and solar irradiance can determine the *fundamental*, upper boundary of photosynthesis; however, the *realised*, effective rate of photosynthesis is limited by biotic and abiotic interactions, such as competition for space and the availability of essential nutrients. These limitations are amplified when scaled up to subsequent trophic levels when photosynthetic materials are transferred and assimilated into consumer biomass.

These complex interactions, scaling up from solar energy to fish productivity, likely explains the dichotomy in these effects across the growth of reef fishes. Morais and Bellwood (2018; ref. 68) revealed that the relative effect of temperature on the growth of

fishes is much greater than primary productivity, likely owing to the direct ties of temperature to the metabolic ecology of fishes. We therefore included Figure 4 to directly illustrate the metabolic ceiling of community productivity with regards to temperature. We do agree, as suggested, that it would be valuable to visualise the latitudinal gradients in productivity, so we have included a supplemental figure (Fig. S4) showing this relationship.

I like to see big picture studies like this published. It also gives us a better sense of deviations in recent decades due to human alterations of our biosphere. I hope the authors can address my concerns.

RESPONSE: Thank you again for your comments and feedback on our work. We believe these suggestions have greatly improved our manuscript.

Specific comments:

Line 20: "Biodiversity" meaning should be clarified. The term "biodiversity" is a contraction of the phrase "biological diversity" (Wilson and Peter 1988). It was intended to encompass all scales of diversity from genomic to species, populations, communities, ecosystems, and landscapes. The term has been used in some studies as synonymous with species diversity. I know that is not the intent of the authors but a brief clarification here might be useful for readers of this journal.

RESPONSE: We have reworded this sentence to include a brief clarification as suggested (lines 21-23):

High biological diversity (or biodiversity), scaling from genomes to entire ecoregions, has been thought to bolster communities against disturbances, leading to disproportionately higher levels of ecosystem functioning (i.e. the movement or storage of energy/matter).

Line 34: What is included in "ecosystem functioning"? Ecosystem stability? i.e. resistance of fundamental change in structure and function?

RESPONSE: We have included a clarification of our definition of ecosystem function in the aforementioned sentences (lines 21-23). It pertains primarily to the extent of processes, not resilience, resistance, or stability thereof.

Line 72 states: "Here, productivity is defined as the instantaneous biomass accumulated via ontogenetic growth of all individuals (without mortality) over the survey area over the course of one day". Productivity is always a rate so the word "instantaneous" is confusing and unnecessary.

RESPONSE: We agree and have removed "instantaneous".

Line 82: "Hill diversity" is so important to this paper that a reference is warranted.

RESPONSE: We have added the following citations:

Hill, M.O. Diversity and evenness: A unifying notation and its consequences. *Ecology* 54, 427-432 (1973).

Jost, L. Entropy and Diversity. *Oikos* 113, 363-375 (2006).

Line 102: If Hill diversity that emphasizes abundance over species richness is this study's operational definition of diversity, doesn't it skew results towards the title of the study: "Species abundances surpass richness effects in global biodiversity-ecosystem function relationships"

RESPONSE: We chose the title ex post facto of knowing that abundance effects were greater than richness effects. Hill diversity that emphasises abundance effects produced a better-fitting model compared to species richness (as noted on lines 112-117), which is why we retained it as our operational definition of diversity throughout the remainder of the manuscript.

Line 107: Give reference for Hill 1973 here.

RESPONSE: Done.

Line 135: Some colorblind people cannot differentiate these colors.

RESPONSE: Thank you for pointing this out, we have now also added a shape differentiation in the plot.

Line 137: What does "jittered" mean?

RESPONSE: Jittering is used to create a spread in densely plotted points by adding a small amount of random variation. We have now included a brief description, which reads as:

Note the y axis is on the log₁₀ scale and the points have been jittered (i.e. a very small random number has been added or removed) to improve clarity and interpretability of the figure.

Line 153: " species abundances may be the primary underlying driver of productivity across tropical fish communities". This sounds like circular reasoning. You define productivity as mass per time so abundance must be the result not the driver of productivity (note it is the dependent variable in Fig. 4). Why isn't sunlight and low annual temperature variations the drivers of primary and secondary productivity?

RESPONSE: This is an important issue. For clarity, we have reworded this sentence to remove the circularity/confusion surrounding the use of the term "driver" and instead emphasise that abundances are more related to community-level productivity than richness (lines 164-167). The sentence now reads as:

Namely, increases in abundance-emphasised diversity correlated with greater increases in productivity than increases in species richness-emphasised diversity, which suggests that

species abundances, not richness, may be the primary community-level feature giving rise to/or the consequence of highly productive tropical fish communities.

Line 176: Won't changes in per-capita biomass be spatially constrained in warmer water due to increased respiration?

RESPONSE: The relationship between temperature and growth is multifaceted, encompassing both physiological and ecological limitations. While Pauly's GOLT has been proposed as a potential physiological limitation, the strength of evidence to support this theory is currently weak. Indeed, Bigman et al. (2023a) showed that gill surface area, a better predictor than Pauly's original gill area index (see Bigman et al. 2023b), was weakly related to the maximum size and growth of fishes. Even after scaling gill surface area by body size, gill surface area explained very little variation in growth. Other competing hypotheses have been put forth to explain the temperature-size rule across fishes, including Pauly's GOLT and the metabolic costs associated with warmer temperatures, none of which are necessarily mutually exclusive (reviewed by Audzijonyte et al., 2019).

Given these complex physiological and ecological limitations, we have replaced "energetically" with "physiologically" in this sentence and included the reference for Audzijonyte et al. (2019). This now reads as:

Therefore, the shift in configuration of the relationship between abundance and per-capita productivity (i.e. high per-capita productivity and low abundances towards low per-capita productivity and high absolute abundances) is strongest in the tropics, which suggests that the ecological mechanisms underpinning the BEF across marine fish communities (i.e. changes in per-capita biomass production versus abundance) are likely physiologically constrained in warm-water environments^{15,51}.

References

Audzijonyte A, Barneche DR, Baudron AR, Belmaker J, Clark TD, Marshall CT, Morrongiello JR, van Rijn I (2019) Is oxygen limitation in warming waters a valid mechanism to explain decreased body sizes in aquatic ectotherms? *Global Ecology and Biogeography* 28:64–77

Bigman JS, Wegner NC, Dulvy NK (2023a) Gills, growth and activity across fishes. *Fish and Fisheries* 24:730–743

Bigman JS, Wegner NC, Dulvy NK (2023b) Revisiting a central prediction of the Gill Oxygen Limitation Theory: Gill area index and growth performance. *Fish and Fisheries* 24:354–366

Line 194: " In warm, high-diversity systems, species are likely metabolically limited in their capacities to attain larger sizes." That is Pauly's GOLT (gill oxygen limitation theory) and it should be cited here.

RESPONSE: Please see response above.

Reviewer #3:

The manuscript entitled “Species abundances surpass richness effects in global biodiversity-ecosystem function relationships” explored the relationship between biodiversity and ecosystem function (BEF) across over 1000 marine fish species globally. The authors found that species abundance effects were more influential than species richness effects in global BEF relationship. This study applies several statistical models to multiple datasets from literature review, FishBase, and field surveys. The authors did a great job on organizing the analyses and presenting the results. This is a very interesting study, and it could be attractive to a lot of fishery scientists and managers.

RESPONSE: Thank you for your kind and constructive feedback, we believe your comments have greatly improved our manuscript. Please note that line numbers pertain to the track-changed version of the manuscript.

This manuscript is well-written; however, the Methods section may need to be a little more detailed. I have a few concerns about the data collection and statistical models in this study for the authors to consider and/or clarify.

1. The authors mentioned that the fish trait data came from multiple sources (e.g., FishBase or literature review), and they also pointed out that they corrected some recorded errors from FishBase, which is a little bit confusing. If the authors could clearly tell readers how many species were explored from literature review and how many from FishBase, respectively, it will be helpful for readers to better understand the data sources. In addition, it is even better to know what kind of or how many errors exist in FishBase, which will save a lot of effort for the researchers who are using FishBase as well.

RESPONSE: We have included more details surrounding our search protocols (lines 323-328). A total of 5,770 growth curves were derived from FishBase and supplemented by 1,916 curves from the peer-reviewed literature (now specified on lines 323-325). Regrettably, we did not record which growth curves from FishBase were errors, but they were typically typological or rounding errors. We have modified the text accordingly (lines 326-328).

2. The models used to predict Kmax are Extreme Gradient Boosting (XGBoost) models. However, the authors used the term “boosted regression trees (BRT)” in the text. XGBoost and BRT are quite different in model fitting and hyperparameter setting. Here I feel “XGBoost” is a better term to use.

RESPONSE: Thank you. We have changed all instances of BRT to XGBoost.

3. The authors did not provide the von Bertalanffy growth equation, but instead, they directly talked about using XGBoost models to predict growth, which seems a little confusing. I feel it

would be better if the authors could start with writing out the von Bertalanffy equation and then derive the equations from it to estimate K_{max} .

RESPONSE: Morais and Bellwood (2018) published an extensive description with multiple equations describing the quantification of K_{max} and its ties to the von Bertalanffy growth equations. Because our current study is already quite analytically heavy (and described as rather dense by Reviewer 2), we have chosen not to restate all the same equations. However, we have added an additional line (lines 314-315) in the Methods to help that reads as:

Please see ref. 68 for a detailed description of the quantification of K_{max} and its associated equations.

References

Morais, R.A. and Bellwood, D.R., 2018. Global drivers of reef fish growth. Fish and Fisheries, 19(5), pp.874-889.

4. In the XGBoost model hyperparameters, the value of subsample is way too small (<0.1). Since there is no corresponding R code attached in this manuscript, I cannot help testing it accordingly. I hope the author may double-check the value of it.

RESPONSE: We created a tuning grid where we tested every possible combination of hyperparameters and chose the combination which produced the smallest negative log likelihood. We have now clarified the range of values used in the methods (lines 365-369), which reads as:

We modelled K_{max} using a Gamma loss function and selected hyperparameters using the two-step tuning method from ref.^{38,68}, which involved varying the learning rate (η ; 0.1-0.9), regularising parameter (γ ; 0.1-0.9), maximum tree depth (max_depth ; 5, 10, or 15), and the subsample rate (0.1-0.9).

5. In the K_{max} prediction model validation paragraph, it is great for the authors calculating model's bias. It would be better to calculate the estimator's precision as well. The accuracy would be the combination of bias and precision.

RESPONSE: Because K_{max} is a continuous variable, we measured precision as the R^2 of the fitted vs observed values in the test set, which had a median value of 0.72 (range: 0.65-0.76). We have now included "precision" in the sentence (line 382).

6. In the Calculating productivity paragraph, please add and then explain the formula to calculate productivity.

RESPONSE: We have included a simplified equation and description on lines 393-395.

Some minor issues are listed below.

Line 53: Please move number “17” into the parenthesis.

RESPONSE: Reference 17 pertains to the entire statement whereas the references in the parenthesis are contrary to that statement. We have moved the reference to occur before the parenthesis to avoid confusion. Note, this has now become reference 15 in the revision.

Line 101: Please explain leave-one-out information criterion (LOOIC).

RESPONSE: Done. This sentence now reads as:

The abundance index was the least correlated with all other Hill diversity metrics and PC1 (Fig. S2) and, based on leave-one-out information criterion (i.e. information criterion using a cross-validation procedure; LOOIC), generated the best-fitting BEF relationship (Table S2).

Line 222: Please capitalize “f” in “BEf”.

RESPONSE: Done.

Line 311: Both “regression tree” and “regression trees” were used in the text, please keep them consistent.

RESPONSE: As per the previous comment, we have changed all instances of “regression tree” to “Extreme Gradient Boosting models” throughout the text.

Line 348: Please explain “r2”. It was written “R2” in Figure S4. Please be consistent.

RESPONSE: Thank you for pointing out our inconsistencies, we changed it to R^2 in the main text. We also added a definition that R^2 is a goodness of fit (line 376), which now reads as:

The model performance was also assessed by extracting the R^2 (i.e. the goodness of fit) value from fitting a linear model between $\log(\text{predicted})$ against $\log(\text{observed})$ values from the test set.

Line 352: The sentence “We did not...” is not clear. Please try to reword it.

RESPONSE: We have reworded the sentence (lines 380-384) as follows:

We chose to predict growth using the aforementioned method because our XGBoost models achieved a low median prediction bias of -0.005 (minimum, maximum: -0.01, 0.0008) and a high median precision R^2 of 0.72 (0.65, 0.76; Fig. S4), whereas phylogenetic predictive models (e.g. ref.66) can produce predictions with an accuracy as low as 51%.

Line 445: Please explain “Rhat”.

RESPONSE: Rhat is the scale reduction factor and is a measure of chain convergence. We have now modified the main text to read:

We assessed model fit using posterior predictive checks and simulated residuals from the DHARMA package v.0.4.569 and achieved chain convergence for all estimate parameters (scale reduction factor $R_{hat} < 1.01$) and effective sample sizes were all greater than 1,800 (Table S2).

Line 446: How to detect spatial autocorrelation? Not clear.

RESPONSE: We have rephrased the sentence to read:

Using Moran's I, we detected no spatial autocorrelation in the simulated residuals of any of the models.